# The Transaxillary Route as a Second Access Option in TAVI Procedures: Experience of a Single Centre

**DOI:** 10.3390/ijerph19148649

**Published:** 2022-07-16

**Authors:** Saverio Muscoli, Valeria Cammalleri, Michela Bonanni, Francesca Romana Prandi, Angela Sanseviero, Gianluca Massaro, Marco Di Luozzo, Marcello Chiocchi, Andrea Ascoli Marchetti, Arnaldo Ippoliti, Alessia Zingaro, Gian Paolo Ussia, Francesco Romeo, Pasquale De Vico

**Affiliations:** 1Department of Cardiology, Tor Vergata University of Rome, Viale Oxford 81, 00133 Rome, Italy; v.cammalleri@hotmail.it (V.C.); michelabonanni91@gmail.com (M.B.); francescaromanaprandi@gmail.com (F.R.P.); angelasanseviero192@gmail.com (A.S.); gianluca88massaro@gmail.com (G.M.); diluozzomarco@gmail.com (M.D.L.); gpussia@hotmail.com (G.P.U.); romeocerabino@gmail.com (F.R.); 2Department of Cardiology, University Campus Biomedico of Rome, Via Alvaro del Portillo 200, 00133 Rome, Italy; 3Department of Radiology, Tor Vergata University of Rome, Viale Oxford 81, 00133 Rome, Italy; marcello.chiocchi@gmail.com; 4Department of Vascular Surgery, Tor Vergata University of Rome, Viale Oxford 81, 00133 Rome, Italy; ascolimarchetti@med.uniroma2.it (A.A.M.); ippoliti@uniroma2.it (A.I.); 5Department of Anesthesia, Tor Vergata University of Rome, Viale Oxford 81, 00133 Rome, Italy; alessia.zingaro.md@gmail.com (A.Z.); devico.p@gmail.com (P.D.V.); 6Department of Cardiology, Faculty of Medicine, Unicamillus-Medical University of Rome, Via di Sant Alessandro 8, 00133 Rome, Italy

**Keywords:** TAVI, transaxillary, deep sedation, aortic stenosis

## Abstract

**Background:** The aim of our study was to determine the feasibility and efficacy of transaxillary (TAX) TAVI in patients not eligible for the transfemoral route. **Methods:** This is a retrospective study of a single center. We analysed 262 patients treated with TAVI. In 17 patients (6.5%), the procedure was performed with the TAX approach. Procedural and hospital data, 30-day safety, and clinical efficacy were assessed and compared between the transfemoral and TAX groups. **Results:** In the TAX groups, we found a higher prevalence of men (*p* = 0.001), smokers (*p* = 0.033), and previous strokes (*p* = 0.02). The EUROSCORE II was higher in the TAX group (*p* = 0.014). The success rate of the device was 100%. TAX was associated with a longer procedure time (*p* = 0.001) and shorter median device time (*p* = 0.034) in minutes. Patients treated with TAX had a longer hospital stay (*p* = 0.005) and higher overall bleeding rate (*p* = 0.001). Peripheral neurological complications were more frequent with TAX (*p* = 0.001), which almost completely resolved by 30 days. **Conclusions:** TAX TAVI is safe and effective and should be considered as a second choice when transfemoral TAVI is not feasible due to severe comorbidities.

## 1. Introduction

Transcatheter aortic valve implantation (TAVI) is a minimally invasive technique used in patients affected by severe aortic stenosis (AS) who are considered inoperable or at high risk for surgical aortic valve replacement (SAVR) [1,2].

Data from intermediate-risk patients demonstrated no difference in mortality between TAVI and SAVR [1].

Recently, PARTNER 3 and Evolut Low-Risk trials have established TAVI in low-risk patients as noninferior and even superior for some of the outcomes to SAVR [3,4,5].

A preoperative multislice computed tomography scan (MSCT) is generally performed to assess the valve complex, define the access site, and avoid complications [6,7] (Figure 1). So far, several alternative access routes for antegrade or retrograde TAVI procedures have been described, namely the transfemoral, transaortic, transapical, transcarotid and transaxillary access routes [8,9,10,11]. The transfemoral (TF) access route is considered the gold standard approach for TAVI; however, it can be precluded by unfavorable anatomical characteristics such as calcification, tortuosity, and peripheral artificial graft. The TF approach is usually unsuitable in 10–15% of patients who are candidates for TAVI [12].

While transapical access showed higher mortality and bleeding complication rates compared with TF access, the alternative transaxillary (TAX) approach, demonstrated high technical success rates with favorable safety profiles in selected patients [13].

In the TAX approach, the short course between the axillary artery and aortic annulus allows greater control of the device; the ideal access site is in the first segment of the axillary artery due to the absence of significant side branches. The left axillary artery is preferred over the right due to a more favorable implantation angle (an angle >30° between the annular plane and the horizontal axis is considered a significant limitation for right-sided TAX access) [11]. 

TAX TAVI is associated with low morbidity, early patient mobilization, and short hospitalization times compared with the transapical or transaortic approach [14]. 

Our study aimed to describe the feasibility and efficacy of TAX TAVI when the TF route is not available.

## 2. Materials and Methods

### 2.1. Patient Population 

From January 2012 to March 2017, 262 patients with severe AS and indication for TAVI were consecutively treated at our institution. All patients had severe symptomatic AS and were classified as high risk for SAVR by a multidisciplinary heart team. 

Patients with porcelain aorta, chest radiation, frailty, hostile chest, severe liver disease/cirrhosis, patent internal mammary artery or other critical conduits adherent to the sternum, severe right ventricle disfunction, degenerative neurological disorders, or any contraindication to extracorporeal circulation were ruled out for possible SAVR [15]. Patients were excluded if a reasonable quality or duration of life (<1 year) because of comorbidities was considered unlikely despite valve replacement.

All patients received a preoperative assessment which included a clinical evaluation of the aortic valve and imaging studies taken with transthoracic echocardiography (or transesophageal when necessary). MSCT was used to assess aortic valve size, root morphology, and arterial vascular access anatomy. The assessment of surgical risk was based on the consensus of a multidisciplinary team meeting, using the EUROSCORE II and the STS score. According to clinical and anatomic criteria, patients selected for TAVI were scheduled for the TF, transapical, or TAX approach. In case of contraindications to the TF approach, the TAX was preferred to the transapical approach. The presence of a patent left internal mammary artery graft to a coronary artery was not considered a contraindication to the TAX approach. TAX TAVI was performed on 17 patients (6.5%) when the TF route was contraindicated due to CT scan evidence of severe peripheral arterial disease. We never needed to perform the transapical or transaortic approach. TAX patients were compared with TF patients (*n* = 245). 

The aim of the present study was to assess the safety and efficacy of TAX TAVI when the TF approach is not suitable. Device success, 30-day safety, and clinical efficacy were assessed according to the VARC2 definition in the overall population and specifically in the transaxillary group [16]. 

Device success was defined as the absence of procedural mortality, correct positioning of the device at the proper anatomical location, and good performance of the prosthetic heart valve without mismatch (mean aortic valve gradient < 20 mmHg or peak velocity < 3 m/s, without moderate or severe aortic regurgitation). Early safety (at 30 days) was defined as the absence of all-cause mortality, all stroke (disabling and nondisabling), life-threatening bleeding, acute kidney injury, coronary artery obstruction requiring intervention, major vascular complication, and valve-related dysfunction requiring repeat procedure. Early clinical efficacy was defined as the absence of all-cause mortality, all stroke, hospitalizations for valve-related symptoms or worsening congestive HF, and valve-related dysfunction mean aortic valve gradient ≥ 20 mmHg and/or moderate or severe prosthetic valve regurgitation [16].

We also evaluated, in hospital and at 30 days, peripheral vascular and neurological complications.

Our institutional anesthetic protocol was founded on deep sedation plus local anesthesia. 

Informed consent was obtained for all patients. The study was conducted in accordance with the provisions of the ethics committee of our institute and with the principles of the Declaration of Helsinki.

### 2.2. Transaxillary Technique

All TAX procedures were performed in a hybrid catheterization laboratory by a multidisciplinary team including interventional cardiologists, echocardiographers, cardiac anesthesiologists, and vascular surgeons. The left arm of the patient was abducted and externally rotated to 90° for adequate artery exposure. Using an ultrasound (US) probe (high linear frequency > 10 MHz), the axillary artery, axillary veins, and nerves were identified (Figure 2).

A temporary pacemaker was advanced into the right ventricular apex through the right femoral vein. The axillary artery course was identified while maintaining the arm abducted and externally rotated at 90°. A 6 Fr Pigtail catheter was inserted through the femoral or the right radial artery and positioned in the aortic root. A 3–4 cm skin and deep-tissue incision was performed in the axillary cavity, with particular attention to avoid injury of the brachial plexus. The axillary artery was isolated, and a 7 Fr introducer was advanced over a standard guidewire and through a multipurpose catheter; a standard soft-tip guidewire was exchanged for Super Stiff Amplatz wire (SSAST-1) (Boston Scientific Corp, Boston, MA, USA). A 18 Fr introducer was advanced on the SSAST-1; the aortic valve was crossed using a standard wire and Amplatz Left 1 or 2 catheter (Cordis Corp., Miami Lakes, FL, USA) and exchanged for SSAT-1 wire with a customized, pigtail-shaped, distal tip placed in the left ventricular apex. After aortic balloon valvuloplasty with rapid right ventricular pacing, the TAVI prosthesis was implanted in the native aortic valve. Finally, the introducer was removed, and the site of sheath entry was closed with a Prolene 5-0 suture. Peripheral pulses were confirmed with Doppler and physical examination. Ultimately, drainage was placed into the surgical site for about 48 h.

### 2.3. Anesthetic Management

Anesthetic management was deep sedation (DS) with spontaneous breathing. Patients were monitored by BIS and Smart CapnoLine GuardianTM 02 Microstream (Philips Medical Systems, Eindhoven, The Netherlands). Anesthetic records included: physical condition, medical history, American Society of Anesthesiologists (ASA) risk score assessment, home therapy medications, any adverse events recorded, induction time, and time to discharge. In the preprocedural phase, two peripheral venous accesses and one arterial access were placed. Each patient was placed supine and fitted with nasal cannulae to administer 4 L/min of O_2_ during the TAX procedure at FiO_2_ 40%. The following parameters were recorded: SpO_2_ (oxygen saturation); FiO_2_% (partial pressure of oxygen); IBP (invasive systemic blood pressure) from a transducer connected to the arterial access found in the preoperative phase, which in turn was connected to the monitor and properly reset; HR (heart rate),number of breaths/minute and monitoring was placed with BIS (bispectral index). If necessary, after triple disinfection and sterile field packing with lidocaine, the patient was locally anesthetized and a central venous catheter was placed in the right or left internal jugular vein under ultrasound guidance to infuse drugs and/or high flow fluids or perform a blood transfusion, especially if it was difficult to find venous access in the preoperative phase. During the procedure, the following parameters were monitored and recorded: SpO_2_, FiO_2_%, IBP, HR (with continuous ECG monitoring), number of breaths/minute, and EtCO_2_ (end-expiratory CO_2_ partial pressure) by a properly calibrated device called Microstream Smart CapnoLine Guardian, BIS. At the end of the procedure, SpO_2_, respiratory dynamics (e.g., breathing cessations), number of breaths/minute, EtCO_2_, and hemodynamic stability (IBP, HR, any intraprocedural hypotension, and rhythm fluctuations) were assessed. In case of desaturation (SpO_2_ < 90%) or apnea > 20 s of ventilation, the patient was supported with a face mask until recovery of adequate respiratory drive was achieved. If the patient did not tolerate the procedure or in case of complications, orotracheal intubation was performed with rocuronium (1 mg/kg) and 2% vol Sevoran. The patient’s state of consciousness was assessed both before and after the procedure using the Richmond Agitation-Sedation Scale (RASS). In addition, the time of drug administration was accurately recorded with the respective dosage, onset of action, and offset, and the visual analog scale (VAS) was used to assess the extent of pain after the procedure. During the procedure, spontaneous deep sedation was performed according to the routine method. The dosages of drugs administered to perform deep sedation were midazolam at a dosage of 0.06 mg/kg iv administered once, and diprivan 1% at a dosage of 0.5 mg/kg iv as a single starting dose. The maintenance dose was achieved by the infusion of diprivan 1% at a rate of 4 mL/h iv and Ultiva 50 mcg/mL at a rate of 3.5 mL/h iv. The drugs were periodically titrated according to the anthropometric characteristics of the patient. LA was obtained by subcutaneously injecting lidocaine (maximum dose 4 mg/kg), in proximity to the access site; usually, a longer duration may be achieved using ropivacaine 0.75%. Continuous infusion of norepinephrine 0.1% was started when more than four boluses of vasoconstrictor agent were required. At the end of the procedure, patients were transferred to the cardiac care unit (CCU) for monitoring and appropriate maintenance therapy.

### 2.4. Statistical Analysis 

Descriptive statistics are used to summarize data; continuous variables are expressed and are presented as mean ± standard deviation or medians and interquartile ranges (IQRs), as appropriate. They were compared using Student’s t-test or Mann–Whitney test. Categorical variables are presented as frequencies and percentages and were compared with the Chi-squared test. Differences were considered significant at *p* < 0.05. Statistical analyses were performed using the Statistical Package for Social Sciences, version 26 (SPSS, Chicago, IL, USA).

## 3. Results

### 3.1. Study Population 

In this study, we retrospectively enrolled 17 patients who underwent TAX TAVI in our cardiovascular unit and were compared with 245 patients who underwent TF TAVI. The demographic and clinical characteristics of the two groups are summarized in Table 1. Our TAX population had a mean age of 80.35 ± 9.52 years. Patients were predominantly men (88.2%). All patients had severe symptomatic AS (mean AVA 0.6 ± 0.1 cm^2^; mean aortic gradient 61 ± 4 mmHg; mean peak velocity 4.5 ± 0.5 m/s). The mean EUROSCORE II was 11.27 (7.33–17.5). The mean STS mortality and STS mortality and morbidity were 7.59 (5.74–8.85) and 29.48 (25.81–35.41), respectively. Most patients had the conventional risk factors for coronary artery diseases. The clinical characteristics between the two study groups are compared in Table 1. We observed a statistically significant prevalence in the TAX group compared with in the TF of men (88.2 vs. 45.7%, *p* = 0.001), current smokers (64.7 vs. 35.9%, *p* = 0.033), prior TIA (23.5 vs. 4.5%, *p* = 0.001), prior stroke (17.3 vs. 4.5%, *p* = 0.02). As we expected, the EUROSCORE II was higher in the TAX (11.27) than in the TF (6.83) group (*p* = 0.014) because, as previously described, patients for the TAX approach were excluded from the classic TF procedure due to severe comorbidities. We found a higher prevalence of chronic kidney disease (CKD) in the TF (69.49%) than in the TAX (35.3%) group (*p* = 0.004).

### 3.2. Intraprocedural Management and Outcomes 

In the TAX population, 10 patients (58,8%) received the Medtronic CoreValve prosthesis (26 mm in 1 patient, 29 mm in 4 patients, and 31 mm in 5 patients), 6 patients (35.3%) received the Medtronic Evolut R (29 mm in 5 patients and 34 mm in 1 patient), and 1 patient (5.8%) received the St. Jude Medical Portico (27 mm). 

Device success was obtained in all patients (100% procedural success rate) without significant difference from the TF group. We observed a significant difference in the median procedural time (skin to skin: time interval from axillary artery puncture to the access closure): 150 (IQR 114.5–165.5) minutes, which was comparable to the TF group, which took 71 (IQR 60–86) minutes (*p* = 0.001); the median device time (time interval from delivery catheter system insertion to its removal) was shorter in TAX 2.5 (IQR 1–7) vs. 3.35 (IQR 1–11) (TF) (*p* = 0.034) minutes. We did not observe any significant difference in fluoroscopy time (*p* = 0.718).

### 3.3. Postprocedural and Thirty-Day Outcomes 

The postprocedural outcomes of the study groups are summarized in Table 2. The need for two or more units of red blood cells within 48 h after the procedure was significantly higher in the TAX group, (35.3% vs. 9.4%, *p* = 0.001). Seven patients (41.18%) developed postprocedural complications in the TAX group compared to one patient in the TF group. In particular, peripheral neurologic complications occurred in three TAX patients (17.6%); they developed postoperative sensory and neurological deficits of the upper limb that was involved in the surgical procedure. Patients who developed sensory or neurological deficits underwent follow-up at one and three months to monitor the course of the neurological deficits through electromyography and somatosensory evoked potentials tests. One month after the procedure, two out of the three patients had regression of symptoms with restitutio ad integrum of the affected upper limb sensitivity. At the three-month follow-up for the last patient, the symptoms progressed to a sensory deficit of the whole upper limb, with an additional functional motor deficit affecting the last two fingers of the hand, probably due to ulnar nerve damage. Notably, no case of acute kidney injury (AKI) or pulmonary complications was observed. No differences in pacemaker implantation were observed between the two groups. The overall median of in-hospital stay after the procedure was longer in the TAX patients, while the median of in-CCU stay did not significantly differ. Any significant differences between the two groups were observed after 30 days (Table 3).

## 4. Discussion

TAVI has become a common technique used for the treatment of severe AS. Heart teams routinely perform a multidisciplinary preprocedural assessment to determine the optimal strategy for vascular access, prosthesis type, and anesthesiologic approach. With the expansion of TAVI to an increasingly large population, including low-risk patients, it is critical to establish a second choice to the TF approach. Over the years, TAX TAVI has proven to be an excellent option as a non-TF approach, although there are no RCTs [17]. Retrograde TF access is the most popular approach but may not be feasible in patients with poor femoral vascular access. A valid alternative nowadays is the TAX access route, which we previously described [11]. 

Other access routes, such as the trans-subclavian, direct transaortic, or carotid access, have been virtually abandoned and are now only used for very few cases.

Experienced operators usually adopt the TAX technique after an adequate learning curve. This technique requires surgical exposure of the vessel and complete immobility of the patient, while the operators work in an unusual and uncomfortable position [17]. Nowadays, this technique is facilitated by new, smaller, and dedicated devices, such as the Corevalve ReValving System introducer sheath (Medtronic, CV, Luxembourg), where the outer diameter of the catheter is 15 Fr (Accu-Trak™ stability layer, Medtronic, CV, Luxembourg), and 12 Fr and the outer diameter of the valve capsule is 18 Fr. 

Patients destined for TAX access were previously excluded from the classic TF approach due to severe lower limb pathology; they are therefore burdened by greater fragility. Nevertheless, the TF approach is safe and effective even in low-risk patients.

Therefore, the TAX approach is not considered an alternative to the TF approach, but is the second most effective and safest choice. The imaging rule is crucial for determining anatomical suitability, device selection, and choosing the optimal access site to avoid common complications such as peripheral vascular lesions and anemia [18,19]. 

In our experience, patients eligible for TAX had a higher risk score than the TF approach. We achieved a successful procedure with the TAX technique in 100% of the cases.

Although more patients in the TAX group were smokers, had a previous CAD, and had had a stroke, they were well-matched for a list of comorbidities (Table 1). In particular, equal numbers of patients had dyslipidemia, arterial hypertension, diabetes, COPD, and previous bypass surgery, and yet there was no difference in 30-day outcomes. Furthermore, despite the need for a surgical cutdown for the TAX approach, fluoroscopy time was the same as than in the TF approach (*p* = 0.718).

In the TAX group, procedural time was longer due to exposure and surgical preparation of the access route.

Conversely, we observed a lower device time in the TAX group, likely due to the shortest route and better device control that operators have from this angle. Moreover, the first operator had a comfortable position at the table, away from the X-ray tube.

The TAX route has several advantages, including a relatively straight course of the delivery system from the intersection to the annulus. The straight course of the axillary artery minimizes the bending effect on the delivery system and ensures one-to-one sensitivity of the position of the tip of the delivery system relative to its handle [20].

The patients in whom the TAX approach is chosen are extremely frail and have severe comorbidities that preclude the TF approach. For this reason, selecting an appropriate anesthetic strategy is critical to the procedure’s success.

An alternative is the TAX percutaneous approach, which was first described in 2012. This option reduces bleeding and neurological complications, especially if the axillary artery is not calcified. The percutaneous approach offers an alternative TAX access route, significantly improved by the new closure devices such as ProGlide, which reduces vascular complications [21].

Anesthetic strategies for TAX procedures differ in many centers. The approaches used are general anesthesia (GA) or deep sedation (DS); they are both valid alternatives and can be used depending on the characteristics of the patient and the experience of the surgeon [22].

Despite the advantages of GA, it also has its disadvantages, especially in patients who are considered extremely fragile. By choosing DS, we were able to avoid all the usual complications. 

DS also allowed a short stay in the ICU, especially in 52.9% of TAX patients with a severe form of COPD that would require weaning from mechanical ventilation if GA was used. Although the TAX approach requires a more prolonged procedure, 71 vs. 150 min (*p* 0.001), we did not note any cases of central neurological problems or psychological distress such as transient delirium.

The heart team is crucial to the procedure’s success not only in the preprocedural discussion, but also in post-TAVI management. This study has several limitations: it was a retrospective observational study at a single center, and the sample size was relatively small. The reported outcomes may have been influenced by operator experience. Further studies are needed to clarify the issue.

## 5. Conclusion

Our study showed that TAX is a feasible and well-tolerated alternative approach for TAVI when the TF approach is unsuitable. However, TAX might increase the risk of peripheral neurological complications compared with the TF approach, and a straight follow-up is necessary.

## Figures and Tables

**Figure 1 ijerph-19-08649-f001:**
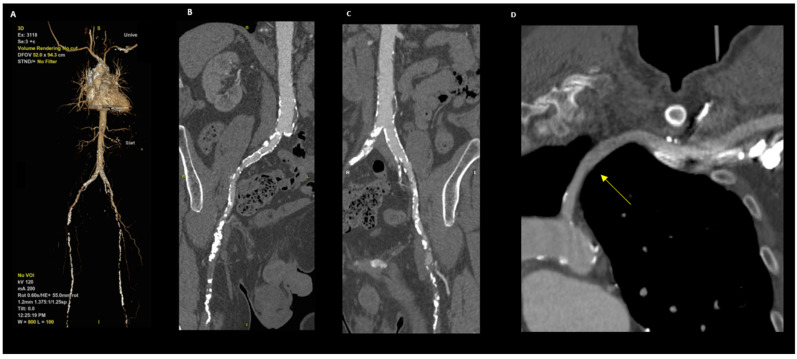
CT angiography for TAVI planning. (**A**) Volume rendering reconstruction shows whole vascular tree with bilateral occlusion of iliac femoral axis. (**B**,**C**) Curved analysis of femoral axis demonstrates occlusion of right femoral artery (**B**) and left femoral artery (**C**). (**D**) Curved analysis of left subclavian artery: evidence of complete patency of vessel.

**Figure 2 ijerph-19-08649-f002:**
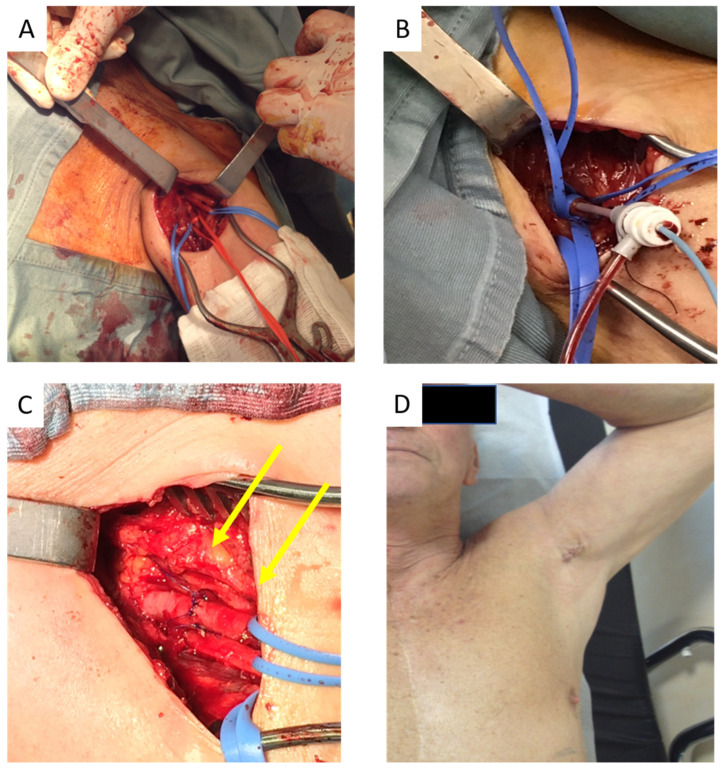
Transaxillary surgical access. (**A**) Median nerve was displaced upward and the axillary vein was displaced downward from the surgical incision; the artery is surrounded by red tape. (**B**) Positioning of the intra-arterial introducer. (**C**) Sutured artery at the end of procedure (yellow arrows indicate the median nerve). (**D**) Postoperative result 10 days after procedure.

**Table 1 ijerph-19-08649-t001:** Demographic characteristics.

	TF Group	TAX Group	*p* Value
	*n* = 245	*n* = 17	
Characteristics of the population			
Age (years), mean ± SD	81.81 ± 6.66	80.35 ± 9.52	0.394
Men, *n* (%)	112 (45.7)	15 (88.2)	0.001
Body surface area (m^2^), mean ± SD	26.34 ± 4.29	24.74 ± 6.31	0.152
Dyslipidemia, *n* (%)	143 (58.4)	12 (70.6)	0.322
Arterial hypertension, *n* (%)	205 (83.7)	16 (94.1)	0.252
Diabetes, *n* (%)	79 (32.2)	3 (17.6)	0.209
Current smoker, *n* (%)	88 (35.9)	11 (64.7)	0.001
Familiarity, *n* (%)	50 (20.4)	5 (29.4)	0.566
History of CAD, *n* (%)	75 (30.6)	10 (58.8)	0.033
CKD	168 (69.49)	6 (35.3)	0.004
COPD	111 (45.3)	9 (52.9)	0.541
CND	21 (8.6)	3 (17.6)	0.210
CABG	31 (12.7)	4 (23.5)	0.202
PTCA	51 (20.8)	7 (41.2)	0.051
Prior MI	46 (18.8)	6 (35.3)	0.099
Prior TIA	11 (4.5)	4 (23.5)	0.001
Prior stroke	11 (4.5)	3 (17.3)	0.020
EF, mean ± SD	49.86 ± 10.86	48.87 ± 9.39	0.611
EUROSCORE II% ± SD	6.83	11.27 (7.33–17.5)	0.014
STS mortality	5.56 (3.42–15.05)	7.59 (5.74–8.85)	0.372
STS mortality and morbidity	23.67 (15.09–45.56)	29.48 (25.81–35.41)	0.237

CNDs: chronic neurological diseases, CAD: coronary artery disease, CKD: chronic kidney disease, COPD: chronic obstructive pulmonary disease, CABG: coronary artery bypass graft, PTCA: percutaneous transluminal coronary angioplasty, MI: myocardial infarction, TAX: transaxillary, TF: transfemoral; TIA: transient ischemic attack, EF: ejection fraction. *p*-value was calculated between transfemoral and transaxillary group.

**Table 2 ijerph-19-08649-t002:** Procedural and in-hospital data.

	TF Group	TAX Group	*p* Value
	*n* = 245	*n* = 17	
Procedural data			
Device success, *n* (%)	238 (97.1)	17 (100)	0.480
Procedural time, min	71 (60–86)	150 (114.5–165.5)	0.001
Fluoroscopy time, min	19.9 (34.5–60)	21(17–25)	0.718
Device time, min	3.35 (1–11)	2.5 (1–7)	0.034
Valve-in-valve, *n*	10 (4.1)	0 (0)	0.845
Balloon postdilation, mean ± SD	24 (9.8)	1 (5.9)	0.917
Overall intraprocedural complication,%	71 (29)	7 (41.2)	0.304
CPR	12 (4.9)	1 (5.9)	0.857
Intraprocedural arrhythmia complications	75 (30.6)	4 (23.5)	0.525
In-hospital data			
In-CCU stay, days mean ± SD	3 (2–4)	3 (2–5)	0.442
In-hospital stay, days	5 (4–7)	7 (6–9)	0.005
Overall postprocedural complication, *n*, (%)	73 (29.8)	7 (41.2)	0.330
Overall vascular complication, *n* (%)	20 (8.2)	1 (5.9)	0.731
Overall bleeding (>2 blood unit), *n* (%)	23 (9.4)	6 (35.3)	0.001
Neurological complications	1 (0.4)	3 (17.6)	0.001
Post-procedural arrhythmia complications	72 (29.4)	4 (23.5)	0.6
Acute kidney injury, *n* (%)	10 (4.1)	1 (5.9)	0.789
Pacemaker implantation, *n* (%)	77 (31.4)	4 (23.5)	0.682

CPR: cardiopulmonary resuscitation, CCU: coronary care unit, AKI: acute kidney injury. *p*-value was calculated between transfemoral and transaxillary group.

**Table 3 ijerph-19-08649-t003:** Thirty-day outcomes (early safety and clinical efficacy), according to VARC-2 definition.

	TF Group	TAX Group	*p* Value
	*n* = 245	*n* = 17	
All-cause mortality, *n* (%)	4 (1.6)	0 (0)	0.595
All stroke, *n* (%)	0 (0)	0 (0)	0.708
Life-threating bleeding, *n* (%)	2 (0.8)	0 (0)	0.708
AKI, *n* (%)	2 (0.8)	0 (0)	
Coronary artery obstruction, *n* (%)	0 (0)	0 (0)	
Major vascular complication, *n* (%)	4 (1.6)	0 (0)	0.595
Repeated procedure for valve dysfunction, *n* (%)	0 (0)	0 (0)	0
Hospitalizations for valve-related symptoms or HF, *n* (%)	2 (0.8)	0 (0)	0.708
NYHA III/IV, *n* (%)	2 (0.8)	0 (0)	0.708
Valve dysfunction, *n* (%)	2 (0.8)	0 (0)	0.708

AKI: acute kidney injury, HF, heart failure; NYHA, New York Heart Association. *p*-value was calculated between transfemoral and transaxillary group.

## Data Availability

Not applicable.

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
