# Peer review of "The Transaxillary Route as a Second Access Option in TAVI Procedures: Experience of a Single Centre"

_ijerph, 2022, doi:10.3390/ijerph19148649_

Round 1

Reviewer 1 Report

It is a retrospective paper reporting transaxillary TAVI outcomes in a single centre (a relatively small-volume centre experience with small number of patients). There is no novel clinical/scientific data/results in this paper as there has been quite a few other published papers reporting axillary approach TAVI.

There are many syntax/formatting issues (eg “trans-axillarian” should be “trans-axillary” in introduction section;   “,” is sued instead of “.” in several places especially in table 1;    “prior ICTUS” would be better stated as “prior stroke” in study population section. The whole second sentence in the conclusion section should be re-written. Also the discussion section should be ideally edited by a native English speaker.

Bleeding requiring two or more units of red blood cells within 48h after the procedure seemed very high (35.3%) in TAX group (eg, in contrast to 4.2% minor bleeding and 0% major or life-threatening bleeding reported in the paper by Zhan Y et al. (Safety and efficacy of transaxillary transcatheter aortic valve replacement using a current-generation balloon-expandable valve. J Cardiothorac Surg 15, 244 (2020). https://doi.org/10.1186/s13019-020-01291-z). Is there any explanation for that?

Reviewer 2 Report

The paper is well written with excellent English. The materials and methods relating to the procedure in question are very detailed and clear, also as regards the anesthetic aspect. The results are easy to read, thanks also to the tables that are also easy to consult. Comments in the discussion are appropriate. The TAX percutaneous approach is mentioned in the discussion; in this regard, I would mention in the text why it was not taken into consideration in your study as an alternative.
